# Long-term carbon sink in Borneo's forests halted by drought and vulnerable to edge effects

Lan Qie et al.

Less than half of anthropogenic carbon dioxide emissions remain in the atmosphere. While carbon balance models imply large carbon uptake in tropical forests, direct on-the-ground observations are still lacking in Southeast Asia. Here, using long-term plot monitoring records of up to half a century, we find that intact forests in Borneo gained 0.43 Mg C ha$^{-1}$ per year (95% CI 0.14–0.72, mean period 1988–2010) in above-ground live biomass carbon. These results closely match those from African and Amazonian plot networks, suggesting that the world's remaining intact tropical forests are now en masse out-of-equilibrium. Although both pan-tropical and long-term, the sink in remaining intact forests appears vulnerable to climate and land use changes. Across Borneo the 1997–1998 El Niño drought temporarily halted the carbon sink by increasing tree mortality, while fragmentation persistently offset the sink and turned many edge-affected forests into a carbon source to the atmosphere.

#A full list of authors and their affiliations appears at the end of the paper

Over the past half-century land and ocean carbon sinks have removed ~55% of anthropogenic $CO_2$ emissions to the atmosphere[1]. At least half this uptake is attributable to the terrestrial biosphere, with multiple independent lines of evidence pointing to the tropics in particular. These include remotely sensed observations of planetary greenness[2], global modelling studies[3], analysis of spatially and temporally resolved aircraft measurements of atmospheric $CO_2$ concentrations, and regional and global transport models[4, 5]. A further body of experimental, observational, and theoretical evidence suggests that a large fraction of these carbon sinks may be driven by rising atmospheric $CO_2$ concentrations[6–8]. Meanwhile, on-the-ground, net carbon gains have been documented in Amazonia and Africa based on long-term observations of nearly 400 ~1 ha plots in structurally intact forests[9–12], and in ten larger plots distributed across the tropics[13]. If these findings apply across the tropics, they imply a mean sink of ≈0.5 Mg C ha$^{-1}$ per year and the annual removal of ≈1.2 Pg C from the atmosphere by tropical forests between 1990 and 2007[14].

Nevertheless, crucial evidence required to establish whether the forest sink is pan-tropical remains missing. Far from the other two large forest blocks in the Amazon and Congo basins, the forests of Southeast Asia are an independent expression of the tropical forest biome, with unique evolutionary history and biota[15, 16]. The largest equatorial rainforests in this region are found in Borneo, where they are typically dominated both in terms of stems and biomass by a single family (Dipterocarpaceae), and characterized by a largely aseasonal climate[16]. While the remaining forests here include the tallest[17], most carbon-dense[18, 19], productive[20] and diverse[15, 19] tropical forests in the world, remarkably little is known about their long-term biomass balance. Trends based on ground measurements have been reported in two studies, but with too few locations monitored—one using eight plots[21], the other three[13]—to draw wider conclusions. Bottom-up estimates of the pan-tropical forest carbon budget either rely on these limited data[6, 22, 23] or simply apply a mean rate from other tropical regions to Asia[14]. Micrometeorological studies using flux towers above the forest canopy suggest Asian

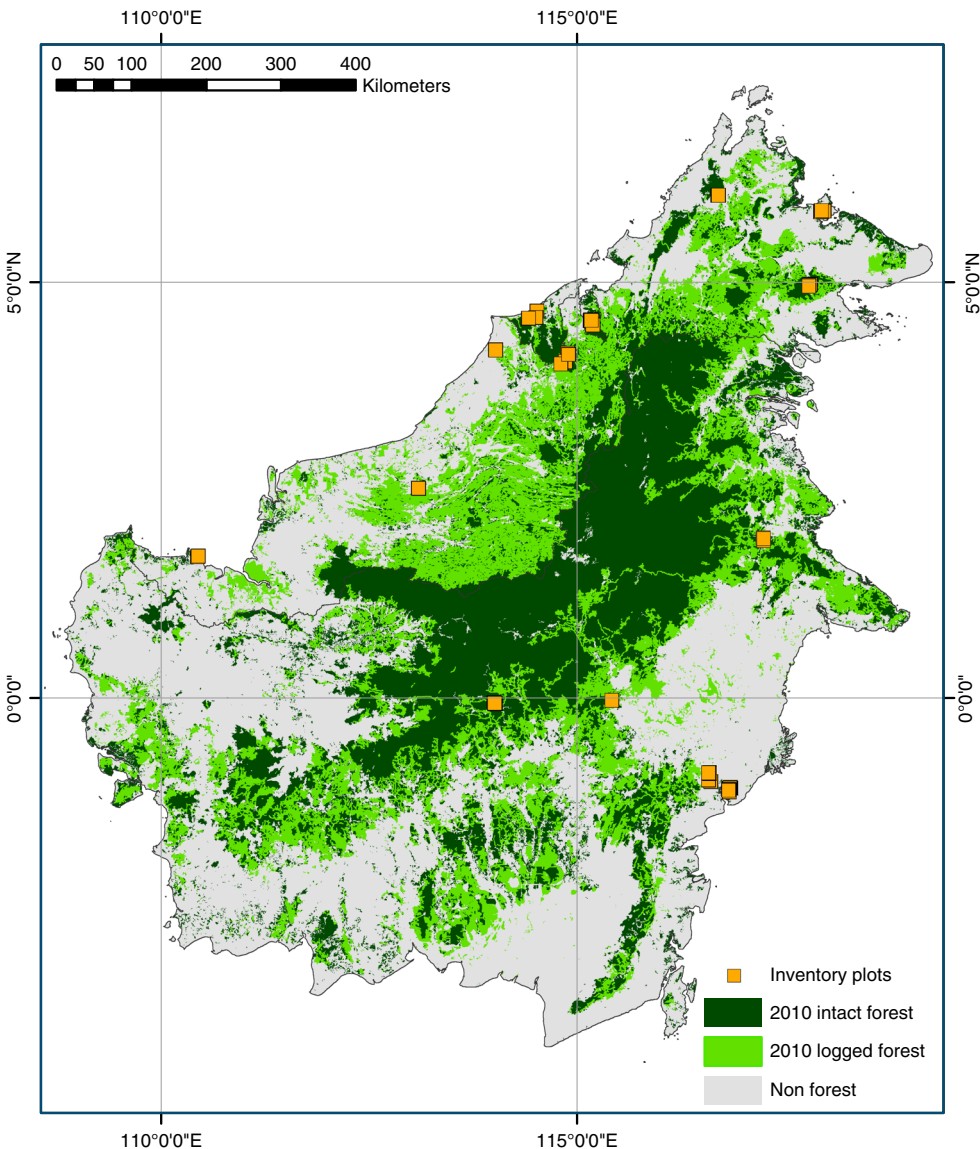

**Fig. 1** Locations of 71 long-term forest monitoring plots shown on the 2010 Borneo forest cover map. Intact and logged forest cover are based on Gaveau et al. 2014[59], accessed at http://www.cgiar-csi.org/portfolio-items/forests-of-borneo. All plots were located in 2010 intact forest areas. Plot symbols overlap, with some obscuring the small forest fragments containing the plots

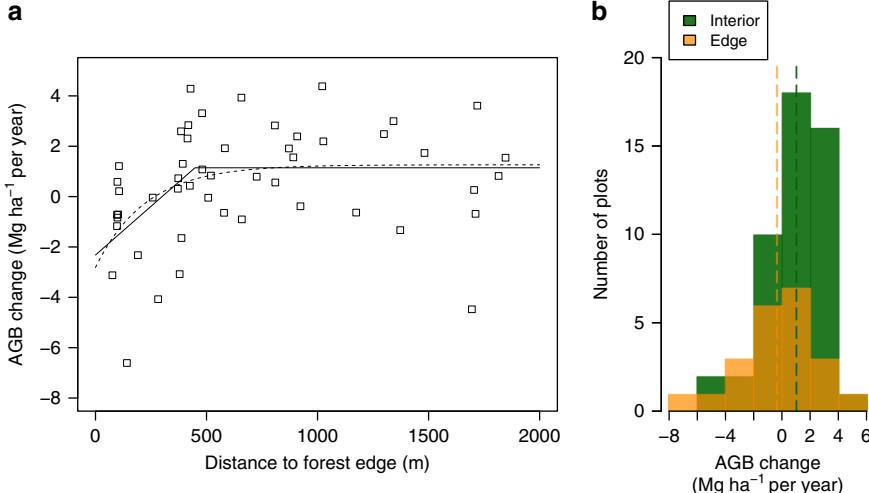

**Fig. 2** Anthropogenic edge impact on plot above-ground live biomass change in structurally intact forests of Borneo. **a** A hockey-stick model (solid line; break-point at 448 m, 1.14 Mg ha$^{-1}$ per year) and asymptotic model (dashed line, asymptote 1.26 Mg ha$^{-1}$ per year) showing saturating edge effect on AGB change. **b** Histograms of plot AGB change, weighted by the cube root of monitoring length, in forest interior (≥448 m, $n = 49$; mean 1.01 Mg ha$^{-1}$ per year, green dashed line) and edge plots (<448 m, $n = 22$; mean −0.36 Mg ha$^{-1}$ per year, orange dashed line)

tropical terrestrial ecosystems are a significant, but highly variable, carbon sink[24]. Meanwhile, top-down assessments based on atmospheric inversions are inconsistent, showing SE Asia to be either a carbon source[25], a sink[26], or carbon neutral[27]. Such top-down assessments estimate net fluxes from all processes, so disaggregation of fluxes to net ecosystem productivity and land use changes is challenging.

Tropical forests are subject to multiple global and regional environmental drivers of change. While a global $CO_2$ driver would favour a sink in the remaining intact forests, at least two other important processes could counteract it in SE Asia. First, SE Asia's tropical forests are among the most fragmented in the world[28, 29], and edge effects may negatively impact biomass accumulation. Neotropical studies show that fragmentation influences ecosystem processes linked to biomass dynamics, increasing tree mortality and recruitment, and altering forest structure and composition, with effects penetrating up to 400 m from forest edges[30–32]. A pan-tropical remote sensing analysis found carbon stock within 500 m of forest edges was on average 25% lower, with reductions extending up to 1.5 km[33], although this likely integrated other anthropogenic effects. A higher-resolution analysis also suggests large, but highly variable, pan-tropical fragmentation emissions of 0.1–0.8 Pg C annually depending on assumptions made about edge loss rates (10–50%) and edge distance (100–300 m)[29]. These highlight the need for better quantification of edge-related losses from tropical landscapes based on long-term observations.

Secondly, severe drought events may offset any sink in living biomass, if they kill enough trees. Two recent severe droughts temporarily reversed the long-term forest carbon sink in the Amazon[11, 34] where forests evolved with annual dry seasons. In Borneo, the largely aseasonal equatorial forests have evolved under a climate regime including El Niño-driven supra-annual droughts, but the frequency and intensity of droughts have increased over recent decades[35]. If trees in Borneo lack adaptation to such increased periodic moisture stress, they may be more vulnerable to drought than in the Amazon[36]. Short-term reports of impacts of the strong El Niño on Borneo's forests in 1982–1983[37] and 1997–1998[13, 38, 39] suggest that extreme droughts may alter forest biomass dynamics, but we lack understanding of the relative magnitude of such short-term impacts relative to long-term trends in these forests.

Here, we develop a temporally and spatially extensive forest dynamics dataset for Borneo. We quantify long-term changes in above-ground live biomass (AGB, dry mass), assess the scale and magnitude of edge effects on forest biomass dynamics, and examine the impact of recent El Niño droughts. Specifically, we hypothesize that: (1) Structurally intact lowland forests in Borneo are gaining AGB, since the most likely driver of the sink in the Amazon and Africa, increasing atmospheric $CO_2$, will also impact Bornean forests. (2) The edges of intact forests will be a reduced carbon sink or a net carbon source to the atmosphere; the greatest effects will be seen closest to the edge and will rapidly diminish with distance. (3) Biomass dynamics in structurally intact Bornean forests were impacted by recent El Niño droughts through increased mortality, but such impacts are short-term and insufficient to cause long-term net loss of biomass.

We assemble data from 71 long-term plots (mean size 1.3 ha) in Borneo's lowland forests (Fig. 1). These have been censused 363 times over the monitoring period spanning 1958–2015 (mean period 1988–2010). Distance from the nearest anthropogenic forest edge was estimated using remotely sensed imagery and published logging road data. For each census and plot we estimated AGB using standard allometric equations[40, 41] based on tree diameters ≥10 cm, wood density[42], and estimated tree height based on locally derived height-diameter models. The mean AGB change that occurred throughout the entire census period was estimated using linear mixed effects (LME) models. We find that Borneo's intact forests have been gaining biomass at a rate comparable to recent estimates for the Amazon and tropical Africa, but this carbon sink is vulnerable to drought and edge effects.

## Results

**Intact forest carbon sink in Borneo.** We first needed to separate edge-affected plots to estimate the interior forest biomass trend. Excluding plots 100 m from the edge and proceeding in 100 m steps to 1000 m from the edge always gave a significant interior sink. The minimum increase in interior AGB was 0.67 Mg ha$^{-1}$ per year with an edge threshold set at 100 m, and generally increased as more edge plots were excluded (Supplementary Table 1). The relationship between anthropogenic edge distance and individual plot AGB change saturates with distance: it was

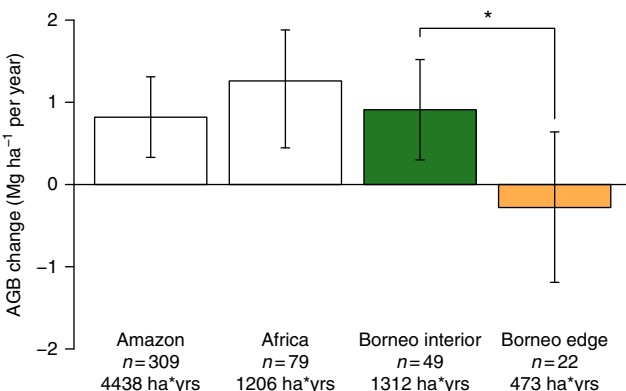

**Fig. 3** Above-ground live biomass change rates in pan-tropical structurally intact forests. Results for Borneo (this study) were based on linear mixed effects (LME) model estimates for the mean period of 1988–2010 (spanning 1958–2015), showing contrast between forest interior plots and edge-affected plots. The most recent published estimates for similar periods for Amazon[10] and tropical Africa[10] are shown for comparison. All estimates are based on direct, ground measurements, with the number of long-term inventory plots and total monitoring effort (in ha years) indicated. Bars are 95% CIs. Only the difference between Borneo interior and edge plots was tested statistically where asterisk indicates significant difference (P = 0.034)

better represented by asymptotic (Akaike's Information Criterion, AIC = 222) and hockey-stick (AIC = 222) models than by either the simple linear (AIC = 227) or null (AIC = 228) models. The hockey-stick model threshold was 448 m (95% bootstrap confidence interval (CI) 393–908 m; Fig. 2a), giving 49 forest interior and 22 edge plots. For this division of plots, using an LME model on plot AGB measurements at every time point, we found that AGB in forest interior plots increased on average by 0.91 Mg ha$^{-1}$ per year (95% CI 0.30–1.52; n = 49) over the monitoring period spanning 1958–2015 (mean period 1988–2010; Fig. 3). This estimate of the interior forest sink equates to a net biomass carbon gain of 0.43 Mg C ha$^{-1}$ per year (0.14–0.72). Our results are not sensitive to the height-diameter allometry used to calculate plot AGB. An alternative commonly used H–D allometry for the SE Asia region[43] gave a similar result (0.93 Mg ha$^{-1}$ per year (95% CI 0.33–1.54)) to our local parameterization.

Additional analyses using alternative approaches to estimating long-term AGB change suggest that the LME approach provides a good but conservative estimate of the interior forest sink. Thus, first, the LME estimate for the forest interior sink is similar to, but smaller than, the plateau value of the hockey-stick model and the asymptote of the asymptotic model of edge effects on AGB change, at 1.14 and 1.26 Mg ha$^{-1}$ per year respectively (Fig. 2a). Second, a non-parametric estimate of AGB change based on the weighted mean of individual plot biomass trends similarly indicates a net forest interior sink of 1.01 Mg ha$^{-1}$ per year (95% bootstrap CI 0.41–1.55). Third, when AGB change is estimated simply as the difference between the long-term mean wood productivity and biomass mortality rates, the mean change is 1.04 Mg ha$^{-1}$ per year (Fig. 4a, b). To explore the effect of outlying values on the estimate of forest interior sink we computed the spectrum of values derived from omitting individual plots and sites one-by-one: no omission altered the significance of our result (Supplementary Fig. 1). AGB gains in forest interior plots were driven by a significant increase in basal area (BA) (Fig. 5a), while functional composition in terms of wood density did not change (Fig. 5b). The measured increase in forest interior biomass was unrelated to initial biomass stocks (Supplementary Fig. 2).

**Edge offset to the carbon sink**. In edge plots, AGB decreased, non-significantly, by −0.28 Mg ha$^{-1}$ per year (95% CI −1.19 to 0.64; n = 22), and this was significantly more negative than the interior plot trend (mean difference −1.19 Mg ha$^{-1}$ per year (−2.29 to −0.09)). The contrast between the forest interior and edge AGB changes was not caused by differences in the rate of production of woody material, which was largely unchanged (7.34 Mg ha$^{-1}$ per year in interior plots vs. 7.52 Mg ha$^{-1}$ per year in edge plots, Fig. 4a). However, the rate of biomass loss due to mortality was non-significantly higher in edge plots than in forest interior plots (7.50 vs. 6.30 Mg ha$^{-1}$ per year, Fig. 4b). Despite losing AGB, the edge plots had positive BA change, though not significantly so (Fig. 5a), but they differed markedly from the interior plots in showing a strong change in tree community floristic composition, with lower wood density taxa gaining (Fig. 5b). Furthermore, in terms of rates of stem dynamics, edge plots had substantially greater turnover than interior plots—both for mortality (2.3% per year in edge plots (2.0–2.5) vs. 1.8% per year (1.6–2.0) in interior plots; Fig. 4d), and for recruitment (2.3% per year (1.9–2.6) vs. 1.6% per year (1.4–1.8); Fig. 4c).

**El Niño drought impact**. All plots experience largely aseasonal precipitation regimes, but are occasionally droughted during strong El Niño events. We could not detect any impact of the 1982–1983 El Niño event on the 17 plots monitored through the event. Notably, AGB mortality appeared unaffected (before 6.43 Mg ha$^{-1}$ per year (4.97–7.89), during 6.24 Mg ha$^{-1}$ per year (4.41–8.07), and after 6.05 Mg ha$^{-1}$ per year (4.71–7.38)). The 1997–1998 El Niño drought, however, significantly altered biomass dynamics of the 19 plots monitored through the event. Before the 1997–1998 drought plot AGB significantly increased (+1.15 Mg ha$^{-1}$ per year (0.10–2.20)), but during drought AGB declined (−2.07 Mg ha$^{-1}$ per year (−4.30 to 0.17)), and then recovered to provide a larger sink (+2.39 Mg ha$^{-1}$ per year (1.09–3.70)) (Fig. 6). Across the entire monitoring periods of these 19 plots, AGB increased by 0.87 Mg ha$^{-1}$ per year (0.03–1.71). The drought associated decline in AGB was driven by a sharp increase in mortality, with AGB mortality averaging 9.05 Mg ha$^{-1}$ per year (6.59–11.51) up from pre-drought levels of 5.49 Mg ha$^{-1}$ per year (4.56–6.42), and not by reductions in wood production, which did not change significantly (before 7.02 Mg ha$^{-1}$ per year (6.44–7.60); during 7.25 Mg ha$^{-1}$ per year (6.43–8.07)). After the 1997–1998 event, while mortality returned to pre-drought levels (5.92 Mg ha$^{-1}$ per year), above-ground wood productivity (AGWP) was significantly higher (8.33 Mg ha$^{-1}$ per year (7.72–8.95)) than pre-drought productivity.

**Discussion**
Our analysis of data from structurally intact forests across Borneo reveals an above-ground live biomass carbon sink of 0.43 Mg C ha$^{-1}$ per year during the late 20th and early 21st centuries (mean period 1988–2010). The sink value is not strongly dependent on the definition of edge distance, and edge effects rapidly diminish with distance (Fig. 2). Our estimate of the intact forest carbon sink is of similar magnitude to recent ground-based estimates for the Amazon[9, 11, 12] and tropical Africa[10] (Fig. 3), and broadly consistent with past results from two large plots in Malaysia[13].

The similarity in both direction and magnitude of net biomass changes across three distinct tropical forest regions suggests that a common driver is causing each to behave in a similar, non-equilibrium way. The new finding from SE Asia contributes significantly to the body of evidence for such a global driver[2–8, 14] as it represents a geographically independent replicate of the ongoing global anthropogenic forcing of planetary atmosphere, climate, and primary productivity. Across the tropics, as

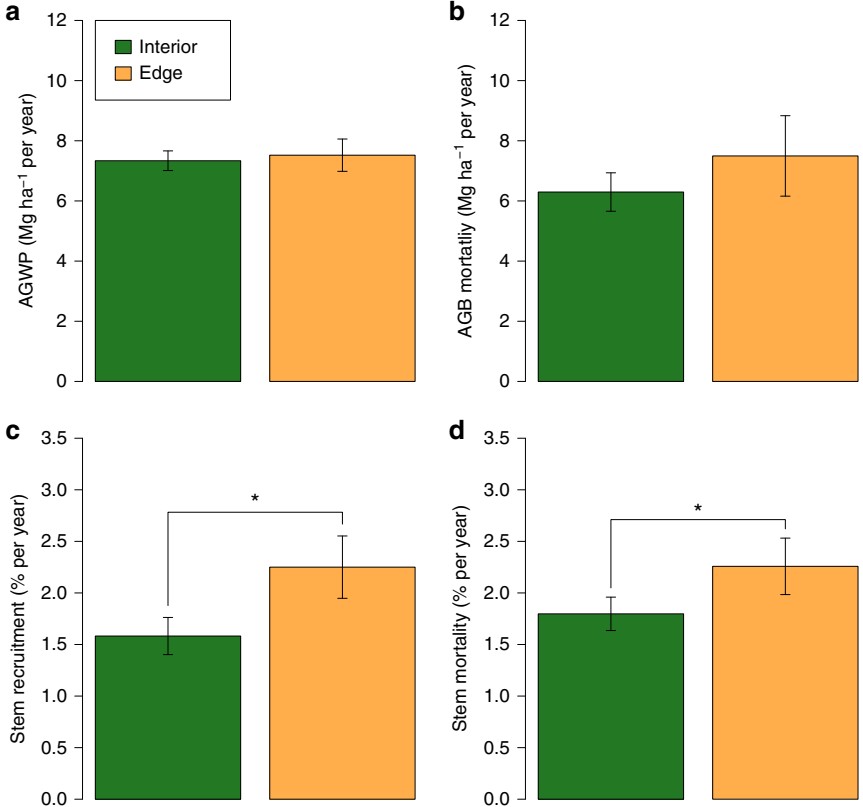

**Fig. 4** Contrasting forest dynamics observed in forest interior and close to anthropogenic edges in Borneo. Above-ground wood productivity (AGWP, **a**), Above-ground live biomass (AGB) mortality (**b**), stem recruitment (**c**) and mortality (**d**) rates are long-term means (mean monitoring period 1988–2010) estimated using linear mixed effects (LME) models. Bars are 95% CIs. Asterisks indicate significant difference (Stem mortality: $P = 0.006$; Stem recruitment: $P < 0.001$). Individual plot values for these variables are presented as histograms in Supplementary Fig. 5 showing variation in both interior and edge plots

elsewhere, there has been a persistent long-term increase in atmospheric carbon dioxide concentrations and in temperature. Carbon dioxide is a key substrate for photosynthesis[6, 7] and the increase in concentration is global and almost uniform geographically, so is hypothesized to lead to a pan-tropical increase in growth[44]. Faster growth is expected to lead to a sink until losses percolate through the system and catch up. Thus, for example, modelling exercises[45] show that a long slow 50-year increase in woody productivity can lead to a forest carbon sink for over a century, until a new dynamic equilibrium at higher biomass is reached. Air temperatures have also increased globally, so could conceivably be an alternative driver of the pan-tropical sink. However, respiration increases with temperature[46], so all else being equal temperature increases are widely expected to decrease tropical tree growth[47], not increase it. Estimates of the net carbon impact of temperature changes at ecosystem-scale for tropical forests nevertheless remain poorly constrained because of substantial technical and practical challenges[44, 48]. How long the sink will continue is not known. Soil nutrients may limit future $CO_2$ fertilization, as seen in temperate zone experiments[49]. N deposition, which has been increasing in the region[50] may be beneficial to plants[2], or may increase soil acidity reducing the availability of limiting soil nutrients[44]. Alternatively, if P is limiting, mechanisms may exist for availability to keep pace with demand[51, 52].

Other, regional and local, processes also shape forest dynamics and may also play a role. Many tropical forests have a long history of human use[53], and individual plots are subject to disturbance and recovery[54]. Our interior forest AGB results could potentially be the consequence of a recovery from a large Borneo-wide past disturbance, thereby leading to a long-term subsequent biomass carbon sink. Here, while we were unable to detect any impact of the 1982–1983 El Niño event, we show that the stronger 1997–1998 event[35, 55] was only responsible for net losses of <2% of total standing biomass and was insufficient to reverse the long-term sink (the 19 plots monitored through the event experienced long-term biomass gain of 0.87 Mg ha$^{-1}$ per year). Droughts exceeding the 1982–1983 and 1997–1998 episodes may have affected Eastern Borneo in the 19th century[35, 56]. However, to attribute the measured biomass increase to long-term recovery a century or more later, the hypothesized disturbance presumably needs to have caused widespread net biomass losses of at least 100 Mg ha$^{-1}$ which is more than an order of magnitude greater than observed in 1997–1998. As well as drought, other historical disturbances may have impacted individual plots. Our dataset includes natural disturbance-recovery dynamics, as all forests at all times are a mosaic of patches at different states of recovery from natural disturbance as an integral part of forest dynamics. While recovering forests become increasingly dominated by heavier-wooded species[57], during our observation window community average wood density did not change (Fig. 5b). Similarly, forests that are recovering substantially tend to start small and gain biomass[58], but our measured rates of net gain are insensitive to initial biomass, with the highest biomass forests having similar net gains to low biomass forests (Supplementary Fig. 2). Overall, the data available appear inconsistent with disturbance-recovery as a major driver of the measured carbon sink in Borneo's intact forests. The close parallels with observations in Amazonia and

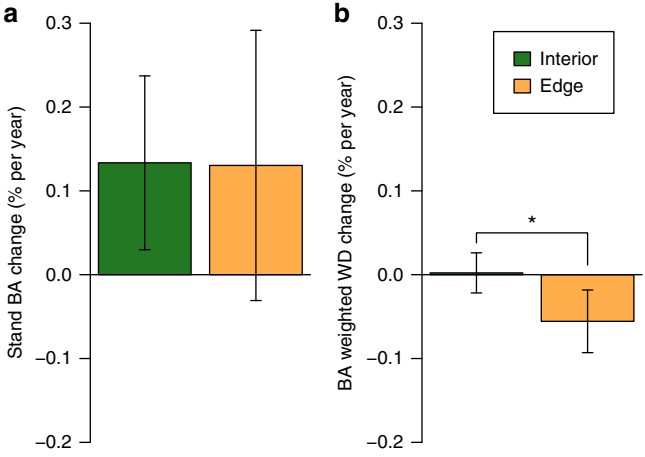

**Fig. 5** Mean changes in stand basal area and basal area weighted mean wood density in forest interior and close to anthropogenic edges in Borneo. Stand basal area (BA, **a**) and BA weighted mean wood density (WD, **b**) were calculated on a proportional basis relative to values of the initial censuses of long-term plots in Borneo. Plots were classified as edge affected if within 448 m from anthropogenic edges. Values represent long-term trends (mean monitoring period 1988–2010) estimated using linear mixed effects (LME) models. Bars are 95% CIs. Asterisk indicates significant difference ($P = 0.011$). Individual plot values for these variables are presented as histograms in Supplementary Fig. 5 showing variation in both interior and edge plots

Africa in terms of the magnitude, sign, and timing of change suggest that a single global driver provides a more parsimonious explanation.

We report changes in above-ground live biomass. However, larger trees, on average, have larger root systems, implying that the whole tree carbon sink is larger than we report. In turn, this might be expected to increase soil carbon storage, but data are lacking on below-ground changes in SE Asian tropical forests. Scaling up from our per-area above-ground carbon sink estimate for structurally intact forests to a wider region requires accurate and consistent land cover classifications. This however remains a challenge. Remaining forests are embedded in a relatively fine-grained mosaic of land uses, with selectively logged forests particularly difficult to distinguish from intact forests using remotely sensed datasets[14, 59], and definitions of 'intact' forest vary in such studies[60, 61]. A detailed analysis of Borneo's forest cover change reported that in 2010 there were 21 Mha of intact forest, defined as unlogged and >700 m from logging roads[59] (Fig. 1). The analysis includes some forest types we lack data for (>1000 m in elevation, peat swamp, fresh water swamp and mangrove) but otherwise lies within our classification of structurally intact forest free from anthropogenic edge effects (Fig. 1). Assuming that all the 21 Mha of intact forest have the same carbon accumulation rate as forests in our study, our observations suggest a total above-ground intact forest carbon sink of ≈9 Tg per year in Borneo.

Logged forest covered an additional 18 Mha of Borneo in 2010[59]. If the drivers of intact forest sink also favour biomass gains in logged forests through similar underlying mechanisms, this would imply a total sink of ≈17 Tg C per year provided by Borneo's 39 Mha of logged and unlogged forest. Larger estimates are possible if logged forests are gaining biomass more rapidly as seen in studies showing rapid carbon stock recovery post logging in Asia[62], the Amazon[63] and Africa[64]. Conversely, any sink would be offset in areas undergoing forest degradation, e.g., adjacent to logging roads, loading yards and skid trails[65]. Reducing this

uncertainty requires both improved resolution in land cover monitoring and better understanding of the interplay between global drivers and local anthropogenic processes, for which our findings may serve to lay the groundwork.

Although we demonstrate that structurally intact forests far from edges in Borneo act as a carbon sink, we show two ways the carbon sink may reverse, one temporarily, due to extreme El Niño droughts, and one over the long-term, due to impacts of anthropogenic edges. We quantified the 1997–1998 drought offset to the biomass sink (−2.07 Mg ha$^{-1}$ per year). More importantly, we find for the first time at the regional scale that there was rapid post-drought biomass recovery in Borneo's forests in the following decade, which returned to being net sinks of similar or greater size than pre-drought, driven by accelerated growth and a return of mortality rate to pre-drought levels (Fig. 6). This demonstrates both the sensitivity (sharp increase in mortality)[36] and resilience (ability to rebound)[66] of the Bornean tropical forests to an extreme drought, similar to the responses observed in Amazonia to the 1997[67] and the 2005[11] droughts. However, recent evidence suggests that repeated droughts may have reduced the biomass recovery capacity in Amazonian forests[34], consistent with observations of a declining Amazon carbon sink[68], so it is possible that the resilience of the Borneo sink may also be challenged in the future.

The offset to Borneo's intact forest carbon sink due to forest fragmentation, however, is clearly ongoing. In contrast to the long-term AGB increase in forest interior, forests closer to anthropogenic edges are more likely to have been a carbon source than a sink. Our data show that forests within 448 m from edges on average lost 0.13 Mg C ha$^{-1}$ per year (0.28 Mg ha$^{-1}$ per year biomass; though not significantly different from zero). The scale of edge effects supported by our data is broadly consistent with the maximum penetration distance of 400 m for a range of ecological processes reported by a comprehensive review of neotropical field studies[32]. Of course edge types, the type of matrix next to the edge and edge age will all affect the temporal trajectories of individual edge forest plots, which require monitoring into the future.

Our analysis also revealed the mechanisms by which edge effects impact forest biomass dynamics. Edge forests may sequester less carbon than forest interior due to (1) lower above-ground wood productivity (AGWP), or (2) higher AGB mortality, or a combination of the two. We find that the biomass decline of edge forests was associated with elevated mortality (both the stem mortality rate and AGB mortality) but with no detectable change in AGWP (Fig. 4). However, elevated mortality in edge-affected forests was compensated for by recruitment, which is also significantly higher than forest interior (Fig. 4). This higher stem turnover near anthropogenic edge forests was accompanied by a significant decrease in stand level mean wood density, unlike in interior forests where WD was unchanged (Fig. 5). That is, near anthropogenic edges forests underwent a compositional shift towards lower wood density species, likely due to increased disturbance. We suspect that additional tree-falls and seed input from nearby disturbed areas have led to a greater probability of recruitment of earlier-successional lower wood density species, leading to the patterns we see.

Overall, our results provide new insight into the pan-tropical carbon sink, its driver(s), its sensitivity and resilience to drought, and the ways in which edge effects may degrade it. They also provide new baseline information to assist in regional carbon accounting, especially in light of the ambitious carbon emission cuts pledged by forest-rich nations such as Malaysia and Indonesia in the wake of the UNFCCC Paris Agreement. For example, the finding of an interior forest carbon sink further highlights the importance of protected forests in the region[69], which may attract

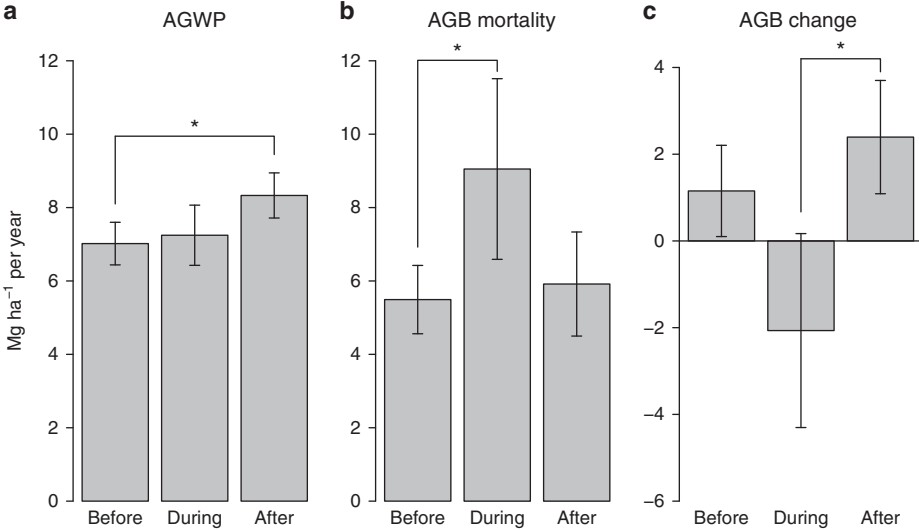

**Fig. 6** Biomass dynamics in 19 forest interior plots in Borneo that were monitored over the 1997–1998 El Niño. Above-ground wood productivity (AGWP, **a**), above-ground live biomass (AGB) mortality (**b**) and AGB change (**c**) were estimated for three mean time intervals: before drought 1978.6–1996.5, during drought 1996.5–2000.0 and after drought 2000.0–2011.1. Values shown are estimated from linear mixed effects (LME) models with 95% CIs (bars) for the 19 plots. Asterisks indicate significant difference based on non-overlapping CIs. Before–during–after trajectories for individual plots for each variable are presented in Supplementary Fig. 6

carbon financing. The high carbon density in SE Asia's forests[18] means that much is at stake. The contrasting biomass trajectories of plots near and far from anthropogenic forest edges indicate that overall biomass dynamics of forests in the region will depend, in part, on the landscape integrity of remaining forests. Based on our estimated rates of AGB change for forest interior vs. edge (0.91 vs. −0.28 Mg ha⁻¹ per year) and the 448 m edge threshold, the minimum area required for a square forest fragment to maintain a carbon sink would be 302 ha (a 1.7 × 1.7 km square). This estimate is likely to be sensitive to edge effect penetration and fragment shape. A worse-case scenario, corresponding to the lower CI (393 m) of our estimated edge threshold but steeper edge offset (Supplementary Table 1) would indicate a minimum fragment area of 719 ha for net carbon sink (a 2.7 × 2.7 km square). On the other hand, if edge effects are mitigated through buffer zone management, thereby preventing biomass loss along reserve boundary, then a forest reserve of any size might serve as a carbon sink. Local studies indicate rather smaller minimum forest fragment sizes to support species of conservation importance (120 ha)[70] or to provide ecosystem services to nearby plantations (200 ha)[71]. While we recognize the value of remnant forests of any size and dimension already existing within anthropogenic matrices, our results suggest that conservation planning needs large forest reserve areas to provide biodiversity and carbon sink co-benefits. We call for urgent protection of remaining tropical forests in SE Asia from further fragmentation, both to preserve the forest interior carbon sink and biodiversity and to halt the increase in edge-related offsets that diminish these key ecosystem services.

## Methods

**Long-term plot data**. Borneo straddles the equator between 4.21°S and 7.04°N, and harbours the largest expanse of lowland rainforests in the Sundaland biogeographical region of SE Asia. All plots in this study were sampled within non-flooded lowland forests (below 1000 m a.s.l.; mixed dipterocarp and kerangas forests) and are geographically well dispersed across the region (Fig. 1, Supplementary Data 1). These forests are on mineral soils that vary in nutrient level[54]. All were mature, structurally intact forests, free from major direct human impacts including fire and logging (plots within logging concessions were in unlogged compartments). Plots had from 2 to 15 censuses (mean 5, median 5), and their monitoring period ranged from 3.8 to 55.8 years (mean 22.6, median 18.3). The combined sampling effort varied over time (Supplementary Fig. 3). Plot size was

between 0.25 and 4.4 ha (area corrected for topography). Plots smaller than 0.4 ha that were within 1 km or less of one another were merged when censuses were synchronous, giving a total of 75 sampling units (referred to as plots hereafter), with a mean size of 1.3 ha (median 0.75). In total our dataset included 95 ha of forest with a combined monitoring effort of 1785 ha years, containing 266,493 measurements on 67,319 individual stems ≥10 cm diameter. Of these individual stems, 96.7%, 94.7% and 78.7% were identified to family, genus and species level respectively, with 22% of the stems lacking species level identification belonging to the large and difficult genus *Syzygium*. All free-standing woody stems with diameter ≥ 10 cm were measured, painted at point of measurement (POM), mapped, and tagged except in seven plots where only trees ≥15 cm were included during census one to five. In these plots we used a minimum diameter of 15 cm throughout, given that in census six to eight, the biomass of 10–14.9 cm trees contributed only ≈3% of total biomass. Plot metadata are summarized in Supplementary Data 1.

Standardized quality control procedures were carried out on diameter records at individual tree level following established protocols[72, 73]. Thus, where potential errors were identified we corrected through extrapolation based on the best available information, i.e., where available, we applied mean growth rate of the same tree for those intervals with accepted measurements, else we applied the growth rate of the same diameter size class from that plot (median growth rate for size class 20–39.9 or >40 cm; mean growth rate for size class 10–19.9 cm). Median growth rate was used for larger size classes because it provides a more robust estimate when sample sizes are small and is more conservative than mean growth rate with respect to the hypothesis of net biomass gain[10, 11]. While the standard POM for diameter was at 1.3 m from the base of the tree, for trees with buttress or deformity at 1.3 m, POM was above it. At censuses where buttress growth might reach existing POM before the next census, a new POM was located sufficiently high above the anticipated future top of buttress and diameter measured at both old and new POMs, providing a taper ratio between both POMs to compute complete diameter series standardized to the old and new POMs[73]. The mean of both diameter series corresponds to an invariant POM located between the old and new (final) POMs, so providing internal consistency over time while using available tree-level information and avoiding biases that result from not accounting for POM change[34, 68, 73]. In cases where a developing buttress already affected the existing POM making it unusable for measurement, a new POM was defined and measured following standard protocols and diameter at the old POM estimated by the quality control protocol. In total, diameter corrections were applied to 2.3% of tree measurements.

**Above-ground live biomass calculation**. Stem above-ground live biomass (AGB, dry mass) was calculated using the allometric equation AGB = 0.0673 × (ρD²H)^0.976 for non-Monocots[40] and AGB = exp(−3.3488 + (2.7483 × ln(D))) for Monocot families[73], where D is stem diameter (in cm), H is height (in m), and ρ is stem wood density (WD, in g cm⁻³) obtained from a global database[42], applying appropriate genus or family-specific values if species level match was not available, and applying mean WD of all identified stems in the plot if no taxonomic information was available for a stem.

Height is an important parameter for estimating tree AGB with allometric equations[43], but is rarely measured directly in tropical inventory plots, and is often estimated from diameter using a $H–D$ model parameterized at continental level[43]. Because this approach does not account for potentially important geographical variation in allometry within continents, we developed $H–D$ models specific to the Sundaland bioregion and four lowland forest types (i.e., combinations of two climate types, 'wet': >3500 mm per year, and 'moist': 1500–3500 mm per year following Chave et al.[74], and two edaphic types, 'mixed dipterocarp' and 'kerangas'), using sub-samples of tree heights actually measured within our plots. Tree heights (total height to tree top) were measured with either a clinometer or laser range finder in the field in 53 of the 71 plots (Supplementary Data 1), with a mean sample size of 100 stems per plot including the 10 largest trees in each plot. For each forest type we selected the best $H–D$ model through comparing three equations, fitted using those trees with measured $H$ and $D$. Firstly, we fitted a Weibull model, $H = a(1 – \exp(–bD^c))$, where $a$, $b$ and $c$ are estimated parameters. Secondly, we fitted the same Weibull model, but using weights proportional to each trees' basal area, to give more importance to large trees during model fitting. Thirdly, we fitted a log–log model, $\ln(H) = a + b(\ln(D))$. We selected the model that minimized prediction error in AGB (i.e., the absolute difference between AGB estimated using measured heights and AGB estimated using heights predicted by the $H–D$ model). In all cases one of the two Weibull models were selected (Supplementary Table 2). To assess the sensitivity of our AGB trend analysis to different $H–D$ models we generated a second set of plot AGB estimates based on the widely used Weibull $H–D$ equation for SE Asia parameterized by Feldpausch et al.[43].

Plot AGB per ha for each census was calculated using the total AGB of live stems. Stand basal area (BA; $m^2$ $ha^{-1}$) and mean WD weighted by stem BA (g $cm^{-3}$) were calculated for each census. At census interval level, above-ground wood productivity (AGWP) of a plot was calculated following Talbot et al.[73] as the sum of AGB gains of surviving and recruit trees; AGB mortality was the summed AGB of trees dying over the interval. Following established procedures[68, 73], AGWP and AGB mortality were corrected to include two small unobserved components relating to trees that die within the census interval: (1) biomass gain and loss of the cohort of unobserved recruits that both enter and die between two successive censuses, and (2) unobserved biomass gain and loss of known trees that die between two successive censuses. For (1), we first estimated the 'true' number of recruits for an interval based on the census-corrected recruitment rate[75], the interval length and the number of surviving stems over that interval (as the base population, see recruitment rate equation below), to derive the number of unobserved recruits. Assuming that the diameter before death of these trees was 10 cm plus growth for one-third of the interval with a mean growth rate of trees in the 10–19.9 cm size class in that plot, we then applied plot mean WD to estimate AGB gain and loss for these unobserved recruits. For (2), we assumed these trees died at the mid-interval, and had grown since last measurement at median growth rate of all trees in the plot of the same size class, to calculate the additional AGB gain and loss for these known dead trees[73]. AGB values were converted to estimates of carbon storage using the mean carbon fraction for tropical angiosperms, of 47.1%[76].

For estimating stem recruitment and mortality rates (% per year), we accounted for the fact that the number of unobserved stems that both enter and die between two successive censuses increases with interval length. Following Lewis et al. 2004[75], we first calculated the observed stem recruitment ($r$) and mortality ($m$) rates for each census interval as $r = 100 \times (\ln(N_s + N_r) – \ln(N_s))/t$ and $m = 100 \times (\ln(N_s + N_m) – \ln(N_s))/t$, where $N_s$ is the number of stems that survived over the census interval, $N_r$ and $N_m$ are the observed number of stems recruited and died over the census interval respectively, and $t$ is census interval length (years). We then estimated the census-corrected rates[75] as $r_{corr} = r \times t^{0.0759}$ and $m_{corr} = m \times t^{0.0759}$.

**Individual plot AGB change estimates**. We first generated simple estimates of AGB change for individual plots, calculated as the slope from a linear regression of AGB against time. Plots with greater sampling effort, in terms of plot area and monitoring length may better represent local AGB changes. Following Lewis et al.[10], an appropriate weighting was determined by examining the deviation of individual plot AGB change rate from the population mean plotted against plot area and monitoring length respectively[10, 11]. For plot area, we found no relationship (Supplementary Fig. 4c), while for monitoring length there was an inverse relationship (Supplementary Fig. 4a). We used cube root of monitoring length as the weight which optimally removed this (Supplementary Fig. 4b, d). Subsequent use of each plot's AGB change metric was based on weighted values.

**Measuring edge distance**. We defined forest edge as the interface of what we considered original forest vegetation and any anthropogenically modified habitat of minimum 1 ha, including active anthropogenic land uses such as inhabited areas, plantations, clear cut logging, as well as regenerating forests from past major anthropogenic disturbances. In selectively logged forests, the exact boundary between unlogged and logged forests was difficult to determine and we considered the nearest logging road as forest edge. We used Google Earth Pro 7.1.7, Landsat imagery from http://earthexplorer.usgs.gov/, and additional GIS data on Borneo's logging roads[59]. Because some plots were near to dynamic anthropogenic

processes, we always selected historical Landsat imagery corresponding to the majority of the plot monitoring period. From each plot's centre point, we searched for the nearest forest edge on remotely sensed imagery using the built-in circle tool of Google Earth Pro, the distance to which was then measured using the software's built-in line tool.

**Establishing edge distance threshold**. To investigate the impact of edge effects without having to define plots as 'edge' or 'interior', we first used an incremental edge distance threshold from 100 m, in 100 m steps, to 1000 m, to generate explorative AGB change estimates for 'edge' and 'interior' plots employing linear mixed effects (LME) models described below. We then tested two models on individual plot AGB change as a function of distance to forest edge: a non-linear asymptotic model $y = a – be^{-cx}$ and a hockey-stick model consisting of two linear segments, the second of which had a zero slope, implemented in the R package SiZer[77]. The asymptotic model is more ecologically meaningful whereas the hockey-stick model is useful in identifying the optimal edge distance threshold. We compared these with a simple linear model (non-asymptotic relationship) and an intercept-only model (no relationship) based on Akaike's Information Criterion (AIC) to test if the hypothesized models better represent the relationship with the given data. We first limited this analysis to plots with edge distance <2,000 m ($n = 50$) because most reported edge effects penetrate only a few hundred metres[30], and then repeated the analysis on all plots to test the robustness of the result. The edge distance threshold identified by the hockey-stick model was then used to divide all plots into two categories, interior and edge plots.

**Modelling AGB dynamics for interior and edge forests**. We used a linear mixed effects (LME) model of AGB observed at each census in each plot as a function of time, a categorical variable indicating whether plots were near an edge, and their interaction. The fixed effect time represents the estimate of AGB change for interior plots, and the time × edge interaction represents how edge effects influence the AGB change. This model formulation thus allows us to quantify the rate of AGB change in interior and edge plots, and test if these rates are different. Plot identity was included as a random effect, allowing us to include any idiosyncratic differences between plots, with a random intercept term capturing variation in AGB between plots and a random slope with the time fixed effect capturing variation in change in AGB among plots[68]. The equation of the model was thus $AGB_{ij} = \beta_0 + \beta_1$ $time_{ij} + \beta_2$ $edge_{ij} + \beta_3$ $time_{ij}$: $edge_{ij} + u_{0i} + u_{1i}$ $time_{ij} + e_{ij}$, where $AGB_{ij}$ is the above-ground biomass in plot $i$ and census $j$, $\beta_0$ to $\beta_3$ are fixed effect parameters, $u_{0i}$ and $u_{1i}$ are respectively the random intercept and slope for plot $i$, and $e_{ij}$ is residual error. The LME model was fitted using the lme function in the nlme R package[78].

An advantage of the LME approach is that it uses all available data and allows us to test all our hypotheses in a single model, providing a powerful solution for unbalanced longitudinal data: individual plots were measured at different times and not all plots were monitored throughout the entire period (Supplementary Fig. 3). However, LME modelling implicitly assumes homogeneity of variance and normally distributed residuals and it is important to check if our model meets these assumptions. LME model validation was done with standard diagnostic plots including examining model standardized residuals against fitted values and plot area. Patterns in LME model residuals suggested that residual variance was correlated positively with fitted value and negatively with plot area. To remove this the model was refitted with weights inversely proportional to variance, where variance = (plot area)$^{-0.208}$ × (fitted value)$^{0.640}$ (estimated during model fitting using the varFunc function in the nlme R package[79]). The resulting model was then validated again before any interpretation was made.

Statistical significance was assessed using the t test statistic. We obtained the 95% CIs of model parameter estimates using a normal approximation to the distribution of the restricted maximum likelihood estimators. Model explanatory power was assessed by calculating marginal R-squared[80] using the MuMIn R package[81]. For AGB, the marginal R-squared was 0.11 (Supplementary Table 3). This value gives the proportion of variation in both space and time explained by model fixed effects; variation in AGB in space was larger than in time, and this spatial variation was primarily accounted for by the plot random effect (Supplementary Table 3).

The same LME model approach was employed to analyse changes in plot variables corresponding to different elements of biomass dynamics, namely plot mean BA and BA weighted plot mean WD. To understand their relative importance, these were calculated on a proportional basis relative to values of the initial censuses in order to apply an equivalent scale to allow comparisons.

We compared the LME model estimate of forest interior AGB change rate with alternatively derived estimates: (1) maximum/asymptotic AGB change predicted from the edge distance threshold analysis, (2) weighted mean of individual plot AGB change in forest interiors, with non-parametric bias-corrected and accelerated bootstrapping to generate 95% CIs (resampling plots 999 times with replacement), and (3) long-term mean AGWP minus AGB mortality, both estimated from LME models similar to above but removing time from fixed effects to generate long-term means.

**Testing for drought impact**. We tested for the potential short-term reversal of the long-term trend associated with impact of the 1982–1983 and 1997–1998 El Niño

droughts on forest biomass using all forest interior plots that were all monitored before, during and after the El Niño events (17 plots for 1982–1983 and 19 plots for 1997–1998). The El Niño time window contained a single census interval from each plot spanning the drought episodes (mean start and end dates for the 1982–1983 window: 1981.4 and 1986.6, and for the 1997–1998 window: 1996.5 and 2000.0). The remaining intervals were assigned to the pre- or post-drought time windows. LME models as described above were then applied for each time window to estimate mean AGWP, AGB mortality, and AGB change.

All analyses were conducted in the R statistical computing environment[82]. Among-site mean level values of AGB change are presented in the text with 95% CIs.

**Data availability**. The plot level biomass dynamics data that support the findings of this study are given in Supplementary Data 1. Tree-by-tree data are curated at www.forestplots.net and additional institutional databases (details given in Supplementary Data 1).

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

## Acknowledgements

This paper is a product of the T-FORCES forest monitoring network (Tropical Forests in the Changing Earth System), supported by an ERC Advanced Grant to O.L.P. and by many institutions, NGOs, government agencies and local communities in Malaysia, Brunei, and Indonesia. We are grateful for historical plot data contributed by the Center for Tropical Forest Science (CTFS; two LAM plots and one BEL plot), the Global Ecosystem Monitoring network (GEM; two LAM plots), Institute for Biodiversity and Environmental Research, Universiti Brunei Darussalam (Brunei plots), Kagoshima University (KIS and KIU plots), Forest Department Sarawak (BKO, LAM, MER and GMU plots), Forest Research Centre, Sabah Forestry Department (SEP plots), the Tropenbos Kalimantan project (ITCI plots), Project Barito Ulu, supported by Indonesian Institute of Sciences (LIPI) (BUL plots), and the STREK project, supported by CIRAD, The Ministry of Forestry of Indonesia, and INHUTANI I (STR plots). We are indebted to a great many individuals who contributed to historical data collection. Contemporary fieldwork was supported by a grant from the ERC (T-FORCES) and from NERC (grants NER/A/S/2000/00532, NE/B503384/1, NE/N012542/1). L.Q. was supported by T-FORCES, CIFOR and NERC NE/P00363X/1. S.L.L. was supported by a Royal Society University Research Fellowship, T-FORCES and a Phillip Leverhulme Prize. O.L.P. is supported by T-FORCES and a Royal Society Wolfson Research Merit Award. M.J.P.S. is supported by T-FORCES and NERC NE/N012542/1. L.F.B. was supported by a NERC studentship to the University of Leeds and a RGS-IBG Henrietta Hutton grant. R.H. was supported by a University of Brunei Darussalam Research Fellowship (2011) and a long-term research project RVO 67985939 from the Czech Academy of Sciences. S.L. received additional support from Primate Conservation Inc. M.S. was supported by a Ministry of Education, Youth and Sports grant of the Czech Republic INGO II LG15051. R.R.E.V. was supported by the Netherlands Foundation for the Advancement of Tropical Research (WOTRO, grant No. W76-217). We thank Forest Department Sarawak, Sabah Biodiversity Centre, Sabah Forestry Department, Forest Department Brunei, Institute for Biodiversity and Environmental Research, University of Brunei Darussalam and Indonesia Ministry of Research, Technology, and Higher Education for research permissions. We thank Bako National Park, Lambir Hills National Park, Gunung Mulu National Park, Kuala Belalong Field Study Centre (KBFSC), Glen Reynolds (SEARRP), Danum Valley Conservation Area, Rainforest Discovery Centre Sepilok, Sepilok Laut Reception Centre, Borneo Orangutan Survival Foundation (BOSF), Sungai Wain Protection Forest Management Unit, WWF East Kalimantan and PT. ITCIKU East Kalimantan for logistical support for fieldwork. We thank Timothy Baker, Roel Brienen, Emanuel Gloor, Adriane Esquivel Muelbert and Nicolas Labrière for comments on the manuscript. We thank our deceased colleagues, John Proctor and Suriantata, for their invaluable contributions to both historical work and our wider understanding of tropical forest ecology.

## Author contributions

O.L.P., Y.M. and S.L.L. conceived the T-FORCES forest monitoring network. O.L.P., S.L.L. and L.Q. conceived and designed the study. L.Q. led the 2013–2015 Borneo field campaigns with the help of O.L.P., Y.M., S.L.L., G.C.P., T.S., K.A.S., K.C.H., L.K.K., H.K., C.M., F.M., E.M., R.N., R.O., C.A.P., R.B.P., E.R., I.S., J.W.F.S., R.S.S., A.T., M.vN., R.R.E.V., I.Y., M.F. and K.H.I. Data contribution was made by S.L.L., P.A., K.A.S., S.A., L.F.B., N.B., F.Q.B., D.B., M.D., S.J.D., G.F., R.H., L.K.K., K.K., S.L., C.M., E.M., L.N., R.N., C.A.P., A.D.P., R.B.P., E.R., P.S., J.W.F.S., R.S.S., M.S., S.T., M.vN., R.R.E.V., I.Y., P.S.K., R.S., L.S.A.L. and M.S.S. O.L.P., S.L.L., M.J.P.S., G.L.G., G.C.P. and W.H. contributed tools to analyse and curate data. G.C.P. produced the forest edge metrics and Fig. 1. L.Q. carried out data analysis. L.Q., S.L.L. and O.L.P. wrote the paper. All co-authors commented on or approved the manuscript.

## Additional information

**Competing interests:** The authors declare no competing financial interests.

Lan Qie [1,2], Simon L. Lewis [1,3], Martin J.P. Sullivan [1], Gabriela Lopez-Gonzalez[1], Georgia C. Pickavance[1], Terry Sunderland[4,5], Peter Ashton[6], Wannes Hubau[7,1], Kamariah Abu Salim[8], Shin-Ichiro Aiba[9], Lindsay F. Banin[1,10], Nicholas Berry[1,11], Francis Q. Brearley[12], David F.R.P. Burslem[13], Martin Dančák[14], Stuart J. Davies[15,16], Gabriella Fredriksson[17,18,19], Keith C. Hamer[20], Radim Hédl[21,22], Lip Khoon Kho[23], Kanehiro Kitayama[24], Haruni Krisnawati[25], Stanislav Lhota[26,27], Yadvinder Malhi[28], Colin Maycock[29], Faizah Metali[8], Edi Mirmanto[30], Laszlo Nagy[31], Reuben Nilus[32], Robert Ong[32], Colin A. Pendry[33], Axel Dalberg Poulsen[33], Richard B. Primack[34], Ervan Rutishauser [35,36], Ismayadi Samsoedin[25], Bernaulus Saragih[37], Plinio Sist[38], J.W. Ferry Slik[8], Rahayu Sukmaria Sukri[8], Martin Svátek[39], Sylvester Tan[40], Aiyen Tjoa[41], Mark van Nieuwstadt[42], Ronald R.E. Vernimmen[43], Ishak Yassir[44], Petra Susan Kidd[45], Muhammad Fitriadi[46], Nur Khalish Hafizhah Ideris[8], Rafizah Mat Serudin[8], Layla Syaznie Abdullah Lim[8], Muhammad Shahruney Saparudin[8] & Oliver L. Phillips[1]

[1]School of Geography, University of Leeds, Leeds LS2 9JT, UK. [2]Department of Life Sciences, Imperial College London, Silwood Park Campus, Ascot SL5 7PY, UK. [3]Department of Geography, University College London, London WC1E 6BT, UK. [4]Center for International Forestry Research, Jl. CIFOR, Situ Gede, Bogor (Barat) 16115, Indonesia. [5]School of Environmental and Marine Science, James Cook University, 1 James Cook Dr, Townsville City, QLD 4811, Australia. [6]Department of Organismic and Evolutionary Biology, Harvard University, 22 Divinity Avenue, Cambridge, MA 02138, USA. [7]Laboratory for wood Biology and Xylarium, Royal Museum for Central Africa, Leuvensesteenweg 13, 3080 Tervuren, Belgium. [8]Environmental and Life Sciences Programme, Faculty of Science, Universiti Brunei Darussalam, Jalan Tungku Link, Gadong BE1410, Brunei Darussalam. [9]Graduate School of Science and Engineering, Kagoshima University, 890-0065 Kagoshima, Japan. [10]Centre for Ecology and Hydrology, Penicuik EH26 0QB, UK. [11]Bioclimate, Thorn House, 5 Rose Street, Edinburgh EH2 2PR, UK. [12]School of Science and the Environment, Manchester Metropolitan University, Chester Street, Manchester M1 5GD, UK. [13]School of Biological Sciences, University of Aberdeen, Cruickshank Building, St Machar Drive, Aberdeen AB24 3UU, UK. [14]Department of Ecology & Environmental Sciences, Faculty of Science, Palacký University in Olomouc, Šlechtitelů 27, CZ-78371 Olomouc, Czech Republic. [15]Center for Tropical Forest Science - Forest Global Earth Observatory, Smithsonian Tropical Research Institute, Washington, DC 20013, USA. [16]Asian School of the Environment, Nanyang Technological University, 50 Nanyang Avenue, Singapore 639798, Singapore. [17]Institute for Biodiversity and Ecosystem Dynamics, University of Amsterdam, 1012 WX Amsterdam, The Netherlands. [18]Pro Natura Foundation, Jl. Jend. Sudirman No. 37, Balikpapan 76112, Indonesia. [19]Pan Eco, SOCP, Jl. Wahid Hasyim No. 51/74, Medan 20154, Indonesia. [20]School of Biology, University of Leeds, Leeds LS2 9JT, UK. [21]Department of Vegetation Ecology, Institute of Botany, The Czech Academy of Sciences, Lidicka 25/27, CZ-60200 Brno, Czech Republic. [22]Department of Botany, Faculty of Science, Palacký University in Olomouc, Šlechtitelů 27, CZ-78371 Olomouc, Czech Republic. [23]Tropical Peat Research Institute, Biological Research Division, Malaysian Palm Oil Board, Bandar Baru Bangi, 43000 Kajang, Malaysia. [24]Graduate School of Agriculture, Kyoto University, Kyoto 606-8502, Japan. [25]Forest Research and Development Center, Research, Development and Innovation Agency, Ministry of Environment and Forestry, Jl. Gunung Batu No 5, Bogor 16610, Indonesia. [26]Department of Animal Science and Food Processing, Faculty of Tropical Agrisciences, Czech University of Life Sciences, Kamýcká 129, 165 00 Praha 6 – Suchdol, Prague, Czech Republic. [27]Ústí nad Labem Zoo, Drážďanská 23, 400 07 Ústí nad Labem, Czech Republic. [28]Environmental Change Institute, School of Geography and the Environment, University of Oxford, Oxford OX1 3QY, UK. [29]International Tropical Forestry, Faculty of Science and Natural Resources, Universiti Malaysia Sabah, Jl. UMS, 88400 Kota Kinabalu, Malaysia. [30]Research Center for Biology, Indonesian Institute of Sciences, Jl. Raya Jakarta-Bogor KM 46, Cibinong 16911, Indonesia. [31]Universidade Estadual de Campinas, Campinas 13083-970, Brazil. [32]Sabah Forestry Department Forest Research Centre, Mile 14 Jl. Sepilok, 90000 Sandakan, Malaysia. [33]Royal Botanic Garden Edinburgh, Edinburgh EH3 5LR, UK. [34]Biology Department, Boston University, 5 Cummington Mall, Boston, MA 02215, USA. [35]Smithsonian Tropical Research Institute, Balboa, Ancon 03092, Panama. [36]Carboforexpert, Hermance 1248, Switzerland. [37]Faculty of Forestry, Mulawarman University, Jl. Pasir Balengkong, 75123 Samarinda, Indonesia. [38]Forests and Societies Research Unit, CIRAD-Univ. Montpellier, Campus International de Baillarguet, TA C-105/D, 34398 Montpellier Cedex 5, France. [39]Department of Forest Botany, Dendrology and Geobiocoenology, Faculty of Forestry and Wood Technology, Mendel University in Brno, Zemedelska 3, 613 00 Brno, Czech Republic. [40]CTFS-ForestGEO Program, Lambir, Miri 98000 Sarawak, Malaysia. [41]Agriculture Faculty of Tadulako University, Jln Soekarno Hatta km 09, Tondo 94118, Indonesia. [42]Utrecht University, Domplein 29, 3512 JE Utrecht, The Netherlands. [43]Deltares, Boussinesqweg 1, 2629 HV Delft, The Netherlands. [44]Balitek-KSDA, Research, Development and Innovation Agency, Ministry of Environment and Forestry, Jl. Soekarno Hatta KM. 38, RT 09 Samboja, Indonesia. [45]Instituto de Investigaciones Agrobiológicas de Galicia (IIAG), Consejo Superior de Investigaciones Científicas (CSIC), Santiago de Compostela 15705, Spain. [46]Sungai Wain Protected Forest Management Unit, KM. 23, Kel. Karang Joang, Balikpapan 76101, Indonesia

