## [Peer Review File · Nature Communications]

Reviewers' comments:

Reviewer #1 (Remarks to the Author):

This an important paper which fills a huge gap in our understanding of the tropical forest carbon sink. The methods, analyses, and interpretation are all excellent. My comments and suggestions below are all minor. My only significant concern is that the implied link between rising CO₂ and increasing biomass is not clearly explained. Is there evidence for increased tree growth in or out of the plots, and why does this increase biomass rather than turnover?

1. There is no mention of data from eddy covariance flux towers in forests. As far as I know, all forest sites in East Asia with towers, including the dipterocarp forest at Pasoh, are sinks.
2. There is no discussion of below-ground biomass. Increasing A-GB does not, by itself, prove the forest is a net sink. Could varying allocation of carbon below ground have influenced the results?
3. A mean observation period of 22 years is not 'long-term' from a forest dynamics perspective. I agree, however, that the pantropical consistency of the results is compelling evidence that the effects are real.
4. The implication is that the increase in biomass is driven by increased tree growth, presumably driven by rising CO₂. However, faster growth does not have a simple relationship with higher biomass, since residence time varies. Increased growth could just lead to higher turnover. A brief explanation of why increased growth is expected to lead to higher biomass over the observation period is needed.
5. What about nutrient limitation, which is likely to be severe in Borneo's deep, old soils? And what about reactive nitrogen deposition, which has been increasing in the region?
6. The tree-ring data does not support increased tree growth across the tropics during this time period (van der Sleen et al. 2017. Trends in tropical tree growth: re-analyses confirm earlier findings. *Global Change Biology*). I realize there is an ongoing debate on how plot and tree-ring data can be reconciled, but I think this at least needs to be mentioned.

Richard Corlett

Reviewer #2 (Remarks to the Author):

The presented study analyzes aboveground biomass changes of tropical forests in Borneo (South East Asia). Using 71 plots with different sizes (between 0.25 ha and 4.4 ha) the authors explore if an increase in biomass could be observed (similar to results for forest plots in Africa and America). In the mean an increase of 0.43 tC per year can be detected. In addition the study investigates biomass changes in the border of forests and could detect a relevant biomass decrease in the first 500 meters. The manuscript shows that two different trends are possible and the sink behavior of forests can be compensated by carbon losses at forest borders.

Introduction and discussion are sound. The manuscript gives a very well overview on the discussion about biomass changes in tropical forests and links their obtained results with the literature. Different to previous papers (e.g. Brienen et al. 2015) is that the authors tackle also impacts of possible disturbances (dynamics at forest borders, El Nino events). The results are novel and interesting, I have a few points concerning used methods and results.

1. The authors stated 'different studies finding the region's lands to be either carbon source, or sink, or carbon neutral' (line 105 -106). It is an interesting question why some studies find different trends compared to this study. I would suggest adding some text to the Discussion section exploring this point (in the context of the used methods).

2. The authors stated that their analysis focus on undisturbed plots (for the forest interior plots). Never the less Fig. S2 shows a large range of different biomass values for the interior plots (AGB). This needs explanation and discussion. Lower biomass values could be also an indication that some of these forests are in earlier successional phase. It is an important question how can this be avoided (to include disturbed forests in the analysis).

3. The observed changes in aboveground biomass are quite small (mean 0.43 Mg C ha for interior plots). Thus the authors should document all details about their fitting methods and weighting concept. Please add the used fitting equation (interaction term, edge variable, e.g. line 452 ff). and document also the quality of fit (e.g. by using r^2 or something similar).

It would be also a helpful for getting an impression on the variability between the plots to add additional graphics showing directly the measured values (like in Fig. S2, compare also Brienen et al. 2015). Results shown in Fig. 4 and 5 should be complemented by graphics in the Appendix showing the derived results at the plot level (AGBP, stem mortality, AGB mortality, stem recruitment, mean wood density, separated for interior and border plots). Similar should be done for Fig. 6.

4. Line 397-398: The derived height-diameter relations should be documented. It should be specified which relations are used for which plots. Are the results different if a more general height-stem diameter relations is used (e.g. from Feldpausch)? This would be interesting to explore.

5. Line 486 (data used for this study): To get a better impression of the variability between the plots, the authors should add additional information from their plots in the supplement file (years of measurements, AGB values as used e.g. Fig . S2, derived and used values for AGBP, mortality, recruitment and wood density per plot).

Additional points

Line 255 -256: 'are widely expected to decrease tropical tree growth', please add more information, there are also papers around that distinguish between temperature effects at leaf and canopy scale (Wood et al. 2012, Biol. Rev.)

Line 346 – 353: I like the calculation of minimum area required assuring that a forest fragment acts as carbon sink (based on the results of this study), this a good point.

Fig. 6: Please add to the legend the time intervals you have chosen for the analysis, or add them into the graphic.

Reviewer #3 (Remarks to the Author):

Review on manuscript for "A long-term carbon sink in Borneo's forests, halted by drought and vulnerable to edges"

General comments:

In this manuscript the authors present data on carbon dynamics in above-ground biomass of tropical forests based on extensive long-term plot monitoring data from 71 plots located across Borneo. Their main findings reveal that intact, pristine forest recently act as carbon sinks potentially driven

by elevated atmospheric CO₂-levels, while these effects are strongly influenced by edge effects occurring after fragmentation and drought events such as in El Niño years.

It was a pleasure to review this manuscript not only as such exhaustive studies done with comparable methodology are rare in such extent, but also as the manuscript is very well written and the results presented in an appealing and comprehensible way. Some of the findings discussed in this paper are not completely novel (as mentioned by Lewis et al 2009, Phillips et al 2017), however results of forest carbon dynamics in this temporal and spatial scope and detail do not exist in Southeast Asia and even rarely anywhere else across the tropics to my knowledge. They are underlining the pan-tropical extent of observed increases in above-ground carbon stocks of intact, pristine forest, while showing that fragmentation and extreme drought events can likely offset this positive effect. Carbon sink and source dynamics and their ongoing changes particularly in such carbon dense areas as found in the wet tropical forest are of global relevance and attract increasing attention not only of scientist in the field, but also of policy makers and the general public. Therefore, I am convinced that these results are an important contribution to the growing field of forest carbon dynamics research and should be considered for publication in Nature Communications after minor adjustments.

I have only some minor critiques and some specific comments below.

I believe that the methods and statistical analyses are sound and described in sufficient detail to make reproduction possible.

While the abstract brings across the main messages of the study, some passages do not sound perfectly smooth. I understand the necessity of keeping it very short, but perhaps the authors tried to squeeze in too many points making the reading flow a bit halting. Particularly the conclusions made in the last two sentences could be disentangled a bit better to make clear which of the statements are derived from these studies' findings (fragmentation and drought effect the carbon sink) and which ones emerged from comparisons to other studies (the sink seems pan-tropical and long-term; land-use plays a role?). It also seems to me that the key role edge effects resulting from fragmentation are playing (which are reflected by the use of 4 figures on edge effect compared to 1 figure on drought) are underrepresented in the abstract.

Specific comments:

- Line 62: Even though the intention of the first sentence becomes clear, the statement sounds a bit awkward as it is now. Which time frame is meant? Do the authors want to say that less than half of annual anthropogenic emissions are actually added to the atmosphere while the other half is directly taken up by oceans and the biosphere again?
- Line 70: It would be fine to say "climate and land-use changes".

- Line 87-88: This seems to me a slightly dangerous comparison to make here. Standing alone this statement may imply that land-use change is not a major issue as the emissions are offset by carbon taken up by the remaining forest. Does the sink counterbalance the projected emissions by ongoing fragmentation and drought effects? How do other natural sources of CO₂ fit in the picture? Furthermore, carbon dynamics in montane tropical forests may react completely different. And as large areas of so far undisturbed forest in Southeast Asia are located in mountainous areas it remains unclear whether the findings can be directly upscaled to all tropical forests.
- Line 107-111: These part is mainly repeating the point made in l. 83-90 and later in l. 99-104 and potentially could be merged.
- Line 210-211: This is a very exciting finding indeed. Could potentially be evaluated in a bit more detail.
- Line 335-336: Here a short ecological perspective on why this may happen could be interesting. For example, it has been shown that forest edges experience a major change in transpiration dynamics effecting tree hydraulics. Therefore, mortality and tree growth performance may be influenced even without an actual disturbance causing shifts in species distribution.
- Line 359: A very important point to highlight.
- Line 368-373: Can any general statement be made on the soil types of the plots, are they mineral soils and were they more or less comparable?

Supplementary Information:

- Page 2, second paragraph: Just from a practical perspective, I would be interested how were the points of diameter measurements marked to find them again after several years? And how often did it occur that “potential errors were identified”?
- Page 3, paragraph 2: How was tree height measured? Using a rangefinder or Vertex? It seems that often particularly tall trees with a wide crown are very difficult to measure in an evergreen closed canopy introducing a bias in the data easily.

Reviewers' comments:

Reviewer #1 (Remarks to the Author):

This an important paper which fills a huge gap in our understanding of the tropical forest carbon sink. The methods, analyses, and interpretation are all excellent. My comments and suggestions below are all minor. My only significant concern is that the implied link between rising CO₂ and increasing biomass is not clearly explained. Is there evidence for increased tree growth in or out of the plots, and why does this increase biomass rather than turnover?

RESPONSE: We have now explained this more fully; see response to point 4 below.

1. There is no mention of data from eddy covariance flux towers in forests. As far as I know, all forest sites in East Asia with towers, including the dipterocarp forest at Pasoh, are sinks.

RESPONSE: Now mentioned. Line 104:

"Micrometeorological studies using flux towers above the forest canopy suggest Asian tropical terrestrial ecosystems are a significant, but highly variable, carbon sink²⁴."

2. There is no discussion of below-ground biomass. Increasing A-GB does not, by itself, prove the forest is a net sink. Could varying allocation of carbon below ground have influenced the results?

RESPONSE: Discussion now added. Changing below-ground allocation cannot affect the reported results because we only report above-ground biomass. However, bigger trees have larger root systems, thus all things being equal, as we see an increase in above-ground tree mass, we likely underestimate the combined above- and below-ground biomass carbon sink in live trees. We have added the following (including a note about pools beyond live trees):

Line 294: "We report changes in above-ground live biomass. However, larger trees, on average, have larger root systems, implying that the live tree carbon sink is larger than we report. In turn, this might be expected to increase soil carbon storage, but data are lacking on below-ground changes in SE Asian tropical forests."

3. A mean observation period of 22 years is not 'long-term' from a forest dynamics perspective. I agree, however, that the pantropical consistency of the results is compelling evidence that the effects are real.

RESPONSE: We agree that 22 years is not very long-term, but it represents from one third to one half of median carbon residence times in tropical forest biomass [Galbraith et al. 2013 Plant Ecology and Diversity].

4. The implication is that the increase in biomass is driven by increased tree growth, presumably driven by rising CO₂. However, faster growth does not have a simple relationship with higher biomass, since residence time varies. Increased growth could just lead to higher turnover. A brief explanation of why increased growth is expected to lead to higher biomass over the observation period is needed.

RESPONSE: We should have been clearer here, details now included. A boost to growth (woody productivity) eventually increases mortality (biomass losses), and hence turnover, but before that new equilibrium is reached there is a carbon sink. This was set out by Chambers et al. 2001 (Nature), Lewis et al. 2004 (Phil Trans B), and is implicit in all global dynamic vegetation models used to model these systems. The discussion (Line 251) now reads:

"Carbon dioxide is a key substrate for photosynthesis^{6,7} and the increase in concentration is global and almost uniform geographically, so is hypothesised to lead to a pan-tropical increase in growth⁴⁴. Faster growth is expected to lead to a sink until losses percolate through the system and catch up. Thus, for example, modelling exercises⁴⁵ show that a long slow 50-yr increase in woody productivity can lead to a forest carbon sink for over a century, until a new dynamic equilibrium at higher biomass is reached."

5. What about nutrient limitation, which is likely to be severe in Borneo's deep, old soils? And what about reactive nitrogen deposition, which has been increasing in the region?

RESPONSE: These are important points, which we did not include as so little is known. We now include a few words on them.

Line 263. "How long the sink will continue is not known. Soil nutrients may limit future CO₂ fertilisation, as seen in temperate zone experiments⁴⁹. N deposition, which has been increasing in the region⁵⁰ may be beneficial to plants², or may increase soil acidity reducing the availability of limiting soil nutrients⁴⁴. Alternatively, if P is limiting, mechanisms may exist for availability to keep pace with demand^{51,52}."

6. The tree-ring data does not support increased tree growth across the tropics during this time period (van der Sleen et al. 2017. Trends in tropical tree growth: re-analyses confirm earlier findings. Global Change Biology). I realize there is an ongoing debate on how plot and tree-ring data can be reconciled, but I think this at least needs to be mentioned.

RESPONSE: Tree-ring studies were initially not mentioned, as the data are few, and fundamentally different - monitoring cohorts, not stands of trees - and there is profound disagreement among tree-ring experts over how to correct for the resulting biases. We do not think covering this here will be helpful because it means bringing in (1) a complex debate and potentially distracting debate about bias-correction techniques for tree-ring analysis, and (2) our judgement having read all sides is that van der Sleen et al. did this incorrectly, and (3) the tree-ring data are extremely few.

However if the editor requests we will insert the following at Line 257:

"Tree-ring studies provide important insights on historical tree-growth. However, bias correction is required in order to convert growth estimates of individual, ageing trees into true stand-level woody productivity. Different approaches give differing sign of long-term trends, making them difficult to compare to our results [Brienen et al. 2017 Global Change Biology; van der Sleen *et al.* 2017 Global Change Biology]."

Richard Corlett

Reviewer #2 (Remarks to the Author):

The presented study analyzes aboveground biomass changes of tropical forests in Borneo (South East Asia). Using 71 plots with different sizes (between 0.25 ha and 4.4 ha) the authors explore if an increase in biomass could be observed (similar to results for forest plots in Africa and America). In the mean an increase of 0.43 tC per year can be detected. In addition the study investigates biomass changes in the border of forests and could detect a relevant biomass decrease in the first 500 meters. The manuscript shows that two different trends are possible and the sink behavior of forests can be compensated by carbon losses at forest borders.

Introduction and discussion are sound. The manuscript gives a very well overview on the discussion about biomass changes in tropical forests and links their obtained results with the literature. Different to previous papers (e.g. Brienen et al. 2015) is that the authors tackle also impacts of possible disturbances (dynamics at forest borders, El Nino events). The results are novel and interesting, I have a few points concerning used methods and results.

1. The authors stated 'different studies finding the region's lands to be either carbon source, or sink, or carbon neutral' (line 105 -106). It is an interesting question why

some studies find different trends compared to this study. I would suggest adding some text to the Discussion section exploring this point (in the context of the used methods).

RESPONSE: Discussion added.

Line 106: "Meanwhile, top-down assessments based on atmospheric inversions are inconsistent, showing SE Asia to be either a carbon source²⁵, or a sink²⁶, or carbon neutral²⁷, largely due to the small number of atmospheric data points driving differing transport models."

2. The authors stated that their analysis focus on undisturbed plots (for the forest interior plots). Never the less Fig. S2 shows a large range of different biomass values for the interior plots (AGB). This needs explanation and discussion. Lower biomass values could be also an indication that some of these forests are in earlier successional phase. It is an important question how can this be avoided (to include disturbed forests in the analysis).

RESPONSE: Explanation and discussion now included. We make clearer that Fig S2 shows that succession is not driving the sink results, as smaller stature plots are not increasing in AGB at higher rates than larger stature plots. Fig 5 also shows that mean wood density is not changing, again showing a lack of succession-driven dynamics. It is also important to note that our dataset includes natural disturbance-recovery dynamics as all forests at all times are a mosaic of patches at different states of recovery from natural disturbance as an integral part of forest dynamics. Thus in the methods we state the plots under study were "mature, structurally intact forests, free from major direct human impacts" (line 393). Idiosyncrasies of each location will account for some of these differences in net AGB change.

The relevant text now reads (line 281):

"As well as drought, other historical disturbances may have impacted individual plots. Our dataset includes natural disturbance-recovery dynamics, as all forests at all times are a mosaic of patches at different states of recovery from natural disturbance as an integral part of forest dynamics. While recovering forests become increasingly dominated by heavier-wooded species⁵⁷, during our observation window community average wood density did not change (Fig. 5b). Similarly, forests that are recovering substantially tend to start small and gain biomass⁵⁸, but our measured rates of net gain are insensitive to initial biomass, with the highest biomass forests having just as high net gains as low biomass forests (Supplementary Fig. 2)."

3. The observed changes in aboveground biomass are quite small (mean 0.43 Mg C ha for interior plots). Thus the authors should document all details about their fitting methods and weighting concept. Please add the used fitting equation (interaction term, edge variable, e.g. line 452 ff). and document also the quality of fit (e.g. by using r^2 or something similar).

RESPONSE: Done: further additional details are now added. We have now expanded the description of the linear mixed effects models, including the model equation and describing in more detail the purpose of each term in the model. We have also clarified that models were weighted by a power function of plot area (as variance in AGB change is greater in smaller plots), and that we used the nlme R package to estimate these weights during model fitting.

Please also note that throughout the results we quantify uncertainty by presenting 95% confidence intervals.

The revised section in Methods (line 476) now reads:

"We used a linear mixed effects (LME) model of AGB observed at each census in each plot as a function of time, a categorical variable indicating whether plots were near an edge, and their interaction. The fixed effect *time* represents the estimate of AGB change for interior plots, and the *time x edge* interaction represents how edge effects influence the AGB change. This model formulation thus allows us to quantify the rate of AGB change in interior and edge plots, and test if these rates are different. Plot identity was included as a random effect, allowing us to include any idiosyncratic differences between plots, with a random intercept term capturing variation in AGB between plots and a random slope with the time fixed effect capturing variation in change in AGB among plots⁶⁹. The equation of the model was thus

$$AGB_{ij} = \beta_0 + \beta_1 \text{time}_{ij} + \beta_2 \text{edge}_{ij} + \beta_3 \text{time}_{ij} : \text{edge}_{ij} + u_{0i} + u_{1i} \text{time}_{ij} + e_{ij}$$

where AGB_{ij} is the above-ground biomass in plot i and census j , β_0 to β_3 are fixed effect parameters, u_{0i} and u_{1i} are respectively the random intercept and slope for plot i , and e_{ij} is residual error. The LME model was fitted using the `lme` function in the nlme R package⁷⁹. Heteroscedasticity was noted during model validation (Supplementary Methods) so to remove this the model was refitted with weights inversely proportional to variance, where variance = (plot area)^{-0.208} x (fitted value)^{0.640} (estimated during model fitting using the `varFunc` function in the nlme R package⁸⁰).

Statistical significance was assessed using the t test statistic. We obtained the 95% CIs of model parameter estimates using a normal approximation to the distribution of the restricted maximum likelihood estimators. Model explanatory power was assessed by calculating marginal R-squared⁸¹ using the MuMIn R package⁸². For AGB, the marginal R-squared was 0.11 (Supplementary Table 4). This value gives the proportion of variation in both space and time explained by model fixed effects; variation in AGB in space was larger than in time, and this spatial variation was primarily accounted for by the plot random effect (Supplementary Table 4).

The same LME model approach was employed to analyse changes in plot variables corresponding to different elements of biomass dynamics, namely plot mean BA and BA weighted plot mean WD. To understand their relative importance, these were calculated on a proportional basis relative to values of the initial censuses in order to apply an equivalent scale to allow comparisons."

It would be also a helpful for getting an impression on the variability between the plots to add additional graphics showing directly the measured values (like in Fig. S2, compare also Brienen et al. 2015). Results shown in Fig. 4 and 5 should be complemented by graphics in the Appendix showing the derived results at the plot level (AGBP, stem mortality, AGB mortality, stem recruitment, mean wood density, separated for interior and border plots). Similar should be done for Fig. 6.

RESPONSE: Done: additional graphs added as new Supplementary Figure 5 & 6 showing plot level variability underlying Figs 4-6.

Individual plot values for variables shown in Fig. 4 (AGWP, AGB mortality, stem mortality, stem recruitment) and Fig. 5 (percentage basal area and wood density change) have been presented as histograms showing variation in both interior and edge plots (Supplementary Figure 5).

For Fig. 6 which shows the AGWP, AGB mortality and AGB change in 19 droughted plots, we present the before-during-after trajectories for individual plots for each variable in Supplementary Figure 6.

4. Line 397-398: The derived height-diameter relations should be documented. It should be specified which relations are used for which plots. Are the results different if a more general height-stem diameter relations is used (e.g. from Feldpausch)? This would be interesting to explore.

RESPONSE: Apologies, this should have been clearer. Additional methods now added, with H-D parameter estimates now added in Supplementary Table 3, and plot AGB using ours and Feldpausch H-D allometry, and new supplementary results. Critically, the AGB change in forest interior plots estimated using Feldpausch H-D allometry is almost identical to ours (Feldpausch: 0.93 Mg ha⁻¹ yr⁻¹ (95% CI 0.33–1.54); Ours: 0.91 Mg ha⁻¹ yr⁻¹ (95% CI 0.30–1.52).

Supplementary Methods reads (SI line 72):

"For each forest type we selected the best *H-D* model through comparing three equations, fitted using those trees with measured *H* and *D*. Firstly, a Weibull function, $H = a(1 - \exp(-bD^c))$, where a , b and c are estimated parameters. Secondly, the same Weibull function, but using weights proportional to each trees' basal area, to give more importance to large trees during model fitting. Thirdly, a log-log model, $\ln(H) = a + b(\ln(D))$. We selected the model that minimised prediction error in AGB (i.e. the absolute difference between AGB estimated using measured heights and AGB estimated using heights predicted by the *H-D* model). In all cases one of the two Weibull models were selected (Supplementary Table 3). To assess the sensitivity of our AGB trend analysis to different H-D models we generated a second set of plot AGB estimates based on the widely used Weibull equation for SE Asia parameterised by Feldpausch *et al.*¹"

The new section in Supplementary Results reads (SI line 119):

"Assessing sensitivity of plot AGB trend to H-D models

A locally parameterised Weibull model was selected to best represent the H-D relation for each of the four forest types (Supplementary Table 3). An alternative commonly used H-D allometry for the SE Asia region¹ gave very consistent results compared to those based on our local parameterisation: forest interior plot AGB increased on average by 0.93 Mg ha⁻¹ yr⁻¹ (95% CI 0.33–1.54; n=49) over the monitoring period spanning 1958-2015 (mean period 1988-2010), and edge affected plot AGB decreased, non-significantly, by –0.17 Mg ha⁻¹ yr⁻¹ (95% CI –1.09 to 0.76; n=22)."

5. Line 486 (data used for this study): To get a better impression of the variability between the plots, the authors should add additional information from their plots in the supplement file (years of measurements, AGB values as used e.g. Fig . S2, derived and used values for AGBP, mortality, recruitment and wood density per plot).

RESPONSE: Done. We have now added all these additional columns in Supplementary Table 2: years or measurement, initial plot AGB, plot mean AGWP, AGB mortality, stem recruitment and mortality rates, and plot mean wood density.

Additional points

Line 255 -256: 'are widely expected to decrease tropical tree growth', please add more information, there are also papers around that distinguish between temperature effects at leaf and canopy scale (Wood et al. 2012, Biol. Rev.)

RESPONSE: Discussion and Wood et al. reference added.

Text in discussion now reads (line 257):

"Air temperatures have also increased globally, so could conceivably be an alternative driver of the pan-tropical sink. However, respiration increases with temperature⁴⁶, so all else being equal temperature increases are widely expected to decrease tropical tree growth⁴⁷, not increase it. Estimates of the net carbon impact of temperature changes at ecosystem-scale for tropical forests nevertheless remain poorly constrained because of substantial technical and practical challenges^{44,48}.."

Line 346 – 353: I like the calculation of minimum area required assuring that a forest fragment acts as carbon sink (based on the results of this study), this a good point.

RESPONSE: Thanks!

Fig. 6: Please add to the legend the time intervals you have chosen for the analysis, or add them into the graphic.

RESPONSE: Done. Apologies this was missed in the previous version.

Reviewer #3 (Remarks to the Author):

Review on manuscript for "A long-term carbon sink in Borneo's forests, halted by drought and vulnerable to edges"

General comments:

In this manuscript the authors present data on carbon dynamics in above-ground biomass of tropical forests based on extensive long-term plot monitoring data from 71 plots located across Borneo. Their main findings reveal that intact, pristine forest recently act as carbon sinks potentially driven by elevated atmospheric CO₂-levels, while these effects are strongly influenced by edge effects occurring after fragmentation

and drought events such as in El Niño years.

It was a pleasure to review this manuscript not only as such exhaustive studies done with comparable methodology are rare in such extent, but also as the manuscript is very well written and the results presented in an appealing and comprehensible way. Some of the findings discussed in this paper are not completely novel (as mentioned by Lewis et al 2009, Phillips et al 2017), however results of forest carbon dynamics in this temporal and spatial scope and detail do not exist in Southeast Asia and even rarely anywhere else across the tropics to my knowledge. They are underlining the pan-tropical extent of observed increases in above-ground carbon stocks of intact, pristine forest, while showing that fragmentation and extreme drought events can likely offset this positive effect. Carbon sink and source dynamics and their ongoing changes particularly in such carbon dense areas as found in the wet tropical forest are of global relevance and attract increasing attention not only of scientist in the field, but also of policy makers and the general public. Therefore, I am convinced that these results are an important contribution to the growing field of forest carbon dynamics research and should be considered for publication in Nature Communications after minor adjustments.

I have only some minor critiques and some specific comments below. I believe that the methods and statistical analyses are sound and described in sufficient detail to make reproduction possible.

While the abstract brings across the main messages of the study, some passages do not sound perfectly smooth. I understand the necessity of keeping it very short, but perhaps the authors tried to squeeze in too many points making the reading flow a bit halting. Particularly the conclusions made in the last two sentences could be disentangled a bit better to make clear which of the statements are derived from these studies' findings (fragmentation and drought effect the carbon sink) and which ones emerged from comparisons to other studies (the sink seems pan-tropical and long-term; land-use plays a role?).

It also seems to me that the key role edge effects resulting from fragmentation are playing (which are reflected by the use of 4 figures on edge effect compared to 1 figure on drought) are underrepresented in the abstract.

RESPONSE: On the abstract, we have reformulated to disentangle the specific results of the study, although we do not have spare words to add further details, unfortunately. We have also made the fragmentation sentence its own point to increase presence in the abstract. The end of the abstract now reads:

"Although both pan-tropical and long-term, the sink in remaining intact forests also appears vulnerable to climate and land-use changes: as across Borneo (i) the 1997-98 El Niño drought temporarily halted it by increasing tree mortality, and (ii) fragmentation persistently offset the carbon sink and turned many edge-affected forests into a carbon source to the atmosphere."

Specific comments:

1. Line 62: Even though the intention of the first sentence becomes clear, the statement sounds a bit awkward as it is now. Which time frame is meant? Do the authors want to say that less than half of annual anthropogenic emissions are actually added to the atmosphere while the other half is directly taken up by oceans and the biosphere again?

RESPONSE: the reviewer is correct, this is what we want to say. As the opening line of the abstract, we are constrained by a strict word limit, hence the lack of timeframe etc., but immediately unpack all of this at line 75, where we say 'past half-century' as the timeframe, and detail the allocation to oceans and land surface. We hope this is a reasonable compromise for readers.

2. Line 70: It would be fine to say "climate and land-use changes".

RESPONSE: Yes, now changed. Thanks.

3. Line 87-88: This seems to me a slightly dangerous comparison to make here. Standing alone this statement may imply that land-use change is not a major issue as the emissions are offset by carbon taken up by the remaining forest. Does the sink counterbalance the projected emissions by ongoing fragmentation and drought effects? How do other natural sources of CO2 fit in the picture? Furthermore, carbon dynamics in montane tropical forests may react completely different. And as large areas of so far undisturbed forest in Southeast Asia are located in mountainous areas it remains unclear whether the findings can be directly upscaled to all tropical forests.

RESPONSE: Thanks, the reviewer is right – without more words the comparison could be misinterpreted. As the comparison to land-use change is not central to the paper we have deleted it.

4. Line 107-111: These part is mainly repeating the point made in l. 83-90 and later in l. 99-104 and potentially could be merged.

RESPONSE: Thanks, we have removed the original line 107-111 to reduce the repetition.

5. Line 210-211: This is a very exciting finding indeed. Could potentially be evaluated in a bit more detail.

RESPONSE: Thanks, we have added more detail. We now include the biomass mortality results from the individual plots, now in added Supplementary Fig 5 and Supplementary Fig 6. We also add biomass mortality values to Supplementary Table 2 for each edge plot.

6. Line 335-336: Here a short ecological perspective on why this may happen could be interesting. For example, it has been shown that forest edges experience a major change in transpiration dynamics effecting tree hydraulics. Therefore, mortality and tree growth performance may be influenced even without an actual disturbance causing shifts in species distribution.

RESPONSE. We have added a sentence on what we think is happening ecologically. Added text reads (line 354):

“We suspect that additional tree falls and seed input from nearby disturbed areas have led to a greater probability of recruitment of earlier-successional lower wood density species, leading to the patterns we see.”

7. Line 359: A very important point to highlight.

RESPONSE: Thanks.

8. Line 368-373: Can any general statement be made on the soil types of the plots, are they mineral soils and were they more or less comparable?

RESPONSE: A general statement on soil of these plots added (line 392). They are mineral soils that vary in nutrient level. Soils don't drive the results in this study because we look at changes within specific stands of trees. The line reads:

“These forests are on mineral soils that vary in nutrient level⁵⁴.”

Supplementary Information:

9. Page 2, second paragraph: Just from a practical perspective, I would be interested how were the points of diameter measurements marked to find them again after several years? And how often did it occur that “potential errors were identified”?

RESPONSE: Trees were painted at the point of measurement; 2.3% of measurements were identified as potential errors. Both points now added to the Supplementary Information (SI line 36 & 42).

10. Page 3, paragraph 2: How was tree height measured? Using a rangefinder or Vertex? It seems that often particularly tall trees with a wide crown are very difficult to measure in an evergreen closed canopy introducing a bias in the data easily.

RESPONSE: Heights were measured with a clinometer or laser range finder; we have added this to the methods. We agree that height measurements are difficult, but we only allow trained scientists to take these measurements, giving us confidence in the results.

REVIEWERS' COMMENTS:

Reviewer #1 (Remarks to the Author):

I am happy with most of the responses to my queries and where I am not entirely convinced I agree the authors have an arguable case. I think this is an extremely valuable study and any further disagreement - on the tree-ring data, for example - should be continued in print. This study is as robust as is possible with the available data and I am not suggesting any more changes.

Reviewer #2 (Remarks to the Author):

The revision of the manuscript has been done carefully. The authors replied well to the raised questions and provided a number of additional and important information which increased the quality of the manuscript. They also revised and clarified numerous formulations.

I enjoyed reading the revised manuscript. I have only a few small points. The paper can be published after minor revision.

Minor comments:

Response letter, page 3, reply to comment 2: 'Idiosyncrasies of each location will account for some of these differences in net AGB change (Fig. S2).'

Thanks for the reply. What do you mean by 'idiosyncrasies'? A specific history of mortality events? Please explain. It would be helpful if you can add a sentence why some of these (mature) plots show very low biomass values (around 250 t/ha, see Fig. S2).

Line 709 : the year of the reference is missing.

Line 712: the year of the reference is missing.

Reviewer #3 (Remarks to the Author):

The suggestions and comments from the previous round were sufficiently addressed by the authors and I have no further objections to the manuscript. I'm convinced that this publication is a valuable contribution to the field of forest carbon dynamics.

REVIEWERS' COMMENTS:

Reviewer #1 (Remarks to the Author):

I am happy with most of the responses to my queries and where I am not entirely convinced I agree the authors have an arguable case. I think this is an extremely valuable study and any further disagreement – on the tree-ring data, for example – should be continued in print. This study is as robust as is possible with the available data and I am not suggesting any more changes.

RESPONSE: We thank the reviewer and agree that tree-ring studies should attract more active research especially in using such data to provide unbiased estimates of historical tree growth.

Reviewer #2 (Remarks to the Author):

The revision of the manuscript has been done carefully. The authors replied well to the raised questions and provided a number of additional and important information which increased the quality of the manuscript. They also revised and clarified numerous formulations.

I enjoyed reading the revised manuscript. I have only a few small points. The paper can be published after minor revision.

RESPONSE: We thank the review for their additional examination of our paper.

Minor comments:

Response letter, page 3, reply to comment 2: 'Idiosyncrasies of each location will account for some of these differences in net AGB change (Fig. S2).'

Thanks for the reply. What do you mean by 'idiosyncrasies'? A specific history of mortality events? Please explain. It would be helpful if you can add a sentence why some of these (mature) plots show very low biomass values (around 250 t/ha, see Fig. S2).

RESPONSE: Indeed, by 'idiosyncrasies' we mean that each forest location has experienced a unique combination of climatic and environmental conditions with a specific history of mortality events. Although it is challenging to provide clear explanation for observed biomass values of particular sites, forests containing patches of earlier successional stages recovering from natural disturbance events generally show lower biomass values. In Fig. S2 we point out this association by saying 'The disturbance-recovery hypothesis predicts that forests at earlier successional stages (corresponding to lower AGB values) will show greater AGB gains.'

Line 709 : the year of the reference is missing.

Manuscript ID: NCOMMS-17-08511B

RESPONSE: Corrected.

Line 712: the year of the reference is missing.

RESPONSE: Corrected.

Reviewer #3 (Remarks to the Author):

The suggestions and comments from the previous round were sufficiently addressed by the authors and I have no further objections to the manuscript. I'm convinced that this publication is a valuable contribution to the field of forest carbon dynamics.

RESPONSE: We thank the review for their additional examination of our paper.